# Antibiotic De-escalation Experience in the Setting of Emergency Department: A Retrospective, Observational Study

**DOI:** 10.3390/jcm10153285

**Published:** 2021-07-26

**Authors:** Silvia Corcione, Simone Mornese Pinna, Tommaso Lupia, Alice Trentalange, Erika Germanò, Rossana Cavallo, Enrico Lupia, Francesco Giuseppe De Rosa

**Affiliations:** 1Department of Medical Sciences, University of Turin, 10126 Turin, Italy; silvia.corcione@unito.it (S.C.); simone.mornesepinna@unito.it (S.M.P.); alice.trentalange@unito.it (A.T.); erika.germano@unito.it (E.G.); enrico.lupia@unito.it (E.L.); francescogiuseppe.derosa@unito.it (F.G.D.R.); 2Tufts University School of Medicine, Boston, MA 02129, USA; 3Infectious Diseases Unit, Cardinal Massaia Hospital, 14100 Asti, Italy; 4Microbiology and Virology Unit, University of Turin, 10126 Turin, Italy; rossana.cavallo@unito.it

**Keywords:** antimicrobial stewardship, de-escalation, emergency department, bloodstream infection, antibiotic treatment

## Abstract

Background: Antimicrobial de-escalation (ADE) is a part of antimicrobial stewardship strategies aiming to minimize unnecessary or inappropriate antibiotic exposure to decrease the rate of antimicrobial resistance. Information regarding the effectiveness and safety of ADE in the setting of emergency medicine wards (EMW) is lacking. Methods: Adult patients admitted to EMW and receiving empiric antimicrobial treatment were retrospectively studied. The primary outcome was the rate and timing of ADE. Secondary outcomes included factors associated with early ADE, length of stay, and in-hospital mortality. Results: A total of 336 patients were studied. An initial regimen combining two agents was prescribed in 54.8%. Ureidopenicillins and carbapenems were the most frequently empiric treatment prescribed (25.1% and 13.6%). The rate of the appropriateness of prescribing was 58.3%. De-escalation was performed in 111 (33%) patients. Patients received a successful de-escalation on day 2 (21%), 3 (23%), and 5 (56%). The overall in-hospital mortality was 21%, and it was significantly lower among the de-escalation group than the continuation group (16% vs 25% *p* = 0.003). In multivariate analysis, de-escalation strategies as well as appropriate empiric and targeted therapy were associated with reduced mortality. Conclusions: ADE appears safe and effective in the setting of EMWs despite that further research is warranted to confirm these findings.

## 1. Introduction

Antimicrobial stewardship (AS) is increasingly recognized as an important multifaceted tool for minimizing unnecessary or inappropriate antibiotic exposure and thereby reducing the rate of antimicrobial resistance (AMR) and associated healthcare costs [1]. AS initiatives strongly promote early de-escalation treatment strategies and thus narrow the spectrum or reduce the number of molecules of an empiric antimicrobial treatment once culture results are available.

Antimicrobial de-escalation (ADE) is a critical aspect of AS programmes. It is strictly dependent on multiple factors, such as the early collection of adequate microbiological samples, pathogen identification, and the administration of an initial anti-infective regimen [2,3]. Several authors have attempted to define ADE from a comprehensive temporal, clinical, biochemical, and microbiological perspective, particularly in the critical care setting [4]. However, there is no universal agreement on the definition and time frame of intervention. Furthermore, in the past, moderately ill patients were more likely to receive ADE than critically ill patients [2]. This practice generated a selection effect that ultimately delayed the incorporation of ADE into evidence-based guidelines throughout hospitals.

Thus, ADE, a key recommendation of the Infectious Disease Society of America’s (IDSA) 2007 stewardship program [5], almost disappeared from the 2016 update [6]. The term de-escalation appears just three times in the entire document but nowhere in a prominent position and is mentioned as a possible metric for evaluating AS programmes. Possible reasons for this lack of emphasis are that ADE is not considered a scientific concept, there is no universally accepted definition of ADE, and that the impact of ADE on different metrics and outcomes, such as mortality, length of hospital stay, and infection recurrence, is unknown. However, in several medical settings, there are usually accepted collectivist norms in the decision-making process about treating infections. These discussions frequently receive input from pharmacists and infectious disease and microbiology specialists and emphasise ADE [7]. 

Nevertheless, several observational studies that focused on ADE in patients admitted to emergency rooms (ERs) and emergency medical wards (EMWs) have reported improved or comparable outcomes with reduced antimicrobial exposure [2]. Similar results have been reported in intensive care unit (ICU) patients [8].

In the ER, the introduction of a sepsis team with the early involvement of infectious diseases consultation (IDC) has been successful in reducing the 14-day mortality. This change also improved the quality of the microbiological work-up, the administration of appropriate antimicrobials, and compliance with the stewardship bundle by reducing the ICU admission rate [9].

Considering the few experiences reported in this setting, the purpose of this study was to examine and describe the prevalence of ADE and the associated factors in a retrospective cohort of patients admitted to a single emergency ward.

## 2. Materials and Methods

A retrospective, observational study of the role of ADE at different times in a single-centre EMW was conducted. This study was part of a more comprehensive AS program. The study was conducted between January 2016 and November 2017 at the City of Health and Science in Turin, Italy. The primary outcome was the rate of clinical and microbiological ADE on days 2, 3, and 5 after admission. Secondary outcomes included factors associated with early ADE, length of stay, and in-hospital mortality. 

Patients were eligible for evaluation if they met all the following criteria: were primarily admitted to the EMW or moved from another ward because of worsening of general conditions; had signs or symptoms suggestive of sepsis or required advanced ventilatory support without an endotracheal tube; had blood cultures (BCs) collected; and were treated with an empirical antibiotic treatment. Demographic data and clinical features were retrieved from the patients’ medical records. For each patient, the quick sequential organ failure assessment (qSOFA) score was calculated on days 1, 3, and 5. Different microbiological samples from other sources were also evaluated in an attempt to establish the source of each patient’s infection. 

If multiple episodes of infection were documented for the same patient during the study period, only the first episode was included. When multiple positive BCs were drawn on different days, only the first positive sample was considered. A single positive BC result out of a multiple set for coagulase-negative staphylococci was considered a contamination, and the sample was excluded from the analysis. The antibiotic treatment was classified as either empiric (ET) or targeted (TT). The rate of appropriate empiric antibiotic treatment (AET), inflammatory biomarkers (procalcitonin, PCT; C-reactive protein, C-RP), and ADE were evaluated according to the BC results and number of days since the BCs were obtained (2, 3, and 5 days after collection). Infections occurring up to 48 h after hospital admission were defined as community-acquired infection (CAI), and those occurring >48 h after admission were considered hospital-acquired infection (HAI). 

ADE was defined as either reduction in the number of antibiotics, reduction of the antimicrobial spectrum, or targeted de-escalation according to the microbiological results. The reasons for ADE were categorized as clinical, independent from the microbiology results and including disappearance or improvement of signs and symptoms of systemic inflammatory-response syndrome; microbiological (also called targeted de-escalation); laboratory biomarker- or IDC-driven. An antimicrobial treatment was defined as microbiologically appropriate if the isolate was susceptible in vitro to ≥1 ET. ADE was retrospectively evaluated and was carried out within EMW by physicians who worked in EMW during the period of the study.

### 2.1. Statistical Analysis

Data were collected in an Excel spreadsheet and analysed using StatView 4.0 (StatView 4.0, JMP software, SAS institute, Cary, NC 27513). Continuous variables are reported as mean (standard deviation) or median (interquartile range). Categorical variables are reported as absolute number (percentage). Nonparametric tests (Wilcoxon, Mann–Whitney, chi-squared, and Fisher’s exact tests) were used for univariate analyses. For categorical variables, chi-squared and Fisher’s tests were used depending on the contingency tables distribution. Non-parametric tests (Wilcoxon and Mann–Whitney) were used for continuous variables and chi-squared and Fisher’s tests for categorical variables. Factors presenting a significant level (*p* < 0.05) at univariate analyses were included in multivariate analyses to assess for risk factors associated with death as an outcome.

### 2.2. Ethics

The study was approved by the Hospital Medical Direction (Protocol No. 0115709). Data were collected in compliance with Italian laws on privacy protection. 

## 3. Results

The study population consisted of 336 patients admitted to EMW, of which 58% (194) were male. The median age of all patients was 70 years (IQR: 60–80). During the preceding six months, 73.8% (248) of patients had at least one previous hospitalization, and half of those (51%) received antibiotics at that time. An active underlying malignancy was recorded in 44% of patients. The mean length of hospital stay was 17 days (IQR: 10–27.5) (Table 1).

Of the 336 BCs collected, 29% (96) were positive, with 8% being polymicrobial. The source of infection was the respiratory tract in 38% of cases, the urinary tract in 22%, intra-abdominal in 21%, and the skin and skin structure in 9%. The majority of infections (73%) were identified as CAI, and 27% were HAI.

Gram-positive organisms were more frequently isolated from BCs than gram-negative organisms (63% vs 34%); S. epidermidis (28%) and S. aureus (25%) were prevalent. Overall, the rate of methicillin-resistance was 13%. Among the gram-negative isolates, E. coli (42%) was the most common, followed by K. pneumoniae (11%). The rate of extended-spectrum beta-lactamases (ESBL)-producing Enterobacteriaceae was 8%, while carbapenemases-producing K. pneumoniae (KPC) was isolated from 6% of BCs; Candida species were isolated from 3% of BCs, mostly C. albicans. An ET was administered to 97% of patients. Fluconazole and caspofungin were the first choices for suspected candidemia (7%).

### The ADE Strategy 

The most frequently prescribed empirical agents were ureidopenicillins (25.1%, *n* = 40), carbapenems (13.6%, *n* = 33), glycopeptides (13.7%, *n* = 44), fluoroquinolones (9.6%, *n* = 31), and third-generation cephalosporins (6.7%, *n* = 16). An initial regimen that combined two agents was prescribed in 54.8% (*n* = 184) of cases. The overall rate of prescription appropriateness was 58.3%, of inappropriateness was 40.0%, and of uncertain appropriateness was 2.7%. Overall, ADE was performed in 33% (111) of the patients.

The ADE rates on days 2 and 3 after the start of ET were 21% and 23%, respectively. Most patients reported a successful ADE at day 5 (56%; *n* = 67). ADE was generally performed according to clinical, microbiological, or biomarker- or IDC-driven strategies, and rates of 76%, 74%, 50%, and 31%, respectively, were reported, although more than one factor influenced the decision. 

Overall discontinuation of antimicrobial therapy until day 5 was 31.5% (*n* = 35) and was performed in 8, 4, and 23 patients, respectively, on day 2, 3, and 5. Moreover, narrowing of antimicrobial spectrum was performed in 53.1% (*n* = 59) of patients collected in this study and was carried out in 7, 16, and 36 patients, respectively, on day 2, 3, and 5

Overall, C-RP was the most commonly used marker of inflammation (80% of cases, of which 87% were on day 1, 84% on day 3, and 75% on day 5), while PCT and beta-D-glucan were available in 50% (67% on day 1, 57% on day 3, and 45% on day 5) and 11% (all performed on day 5) of patients, respectively. Median C-RP values on day 3 were significantly lower in the ADE group than in patients who continued with their original antibiotics (104.04 mg/L vs 138.3 mg/L, *p* = 0.01). PCT was detected in 69%, 54%, and 41% of ADE patients on days 1, 3, and 5, respectively; this was not significantly different from patients who did not de-escalate (67%, 59%, and 47% on days 1, 3, and 5, respectively) (Table 1). Conversely, patients who had lower C-RP levels on day 3 de-escalated more significantly than those with higher values (104 mg/L vs. 138 mg/L; *p* = 0.01). PCT results were excluded from the analysis due to the low number of tests performed. The qSOFA scores on days 2, 3, and 5 were higher in patients who did not de-escalate, although the difference was not significant.

The overall in-hospital mortality rate was 21%, and it was significantly lower among the ADE group than the continuation group (16% vs. 25% *p* = 0.003). The univariate analyses of factors associated with ADE are reported in Table 2.

Either an appropriate ET or TT had a protective effect on mortality (62% vs 44%, *p* = 0.007 and 37% vs. 23%, *p* = 0.006) as well as ADE at any time (34% vs 15%, *p* = 0.013). Multivariate analysis results (Table 3) indicated that appropriate ET and TT and an ADE strategy applied at any time reduced mortality.

Univariate analysis results showed that there were no characteristics associated with ADE strategies. Of note, the qSOFA score was higher in patients who did not de-escalate, but the difference did not reach statistical significance.

## 4. Discussion 

In our study, the overall ADE rate was 33%. The most prescribed empiric antibiotics were ureidopenicillins (25.1%) and carbapenems (13.6%). ADE was performed on day 5 after the start of ET in 56% of patients, on day 3 in 23% of patients, and on day 2 in 21% of patients. ADE was performed by decreasing the number of antibiotics and the spectrum. The overall mortality rate was 21%, and the median in-hospital length of stay was 17 days.

Survival was higher among patients who de-escalated (16% vs. 25%, *p* = 0.003). Multivariate analysis results showed that ADE strategies (*p* = 0.013) and appropriate antibiotic treatment, either empiric (*p* = 0.007) or targeted (*p* = 0.006), were associated with reduced mortality. Our results are in line with other studies on severely ill patients [10,11,12]. The overall rate of methicillin resistance (13%) and multi-drug resistant Enterobacteriaceae (14%) hampered the possibility of ADE and could partially explain the low rate of ADE reported here. 

Interestingly, ADE was performed on day 5 in 56% of patients and within the first three days in 44% of patients, which is when preliminary microbiological data are usually available.

In a clinical setting, the decision to de-escalate a treatment is a multi-layered decision that relies not only on microbiological data but also on clinical stability, source control, and IDC and is definitively a result of a composite evaluation in the EMW. Interestingly, the severity of the illness at the time of admission to the EMW did not influence our decision to change treatment, as the qSOFA scores were not significantly different between the groups. However, patients with negative qSOFA scores tended to de-escalate more frequently than the others. 

C-RP and PCT levels are frequently used as surrogates for clinical response in patients with suspected or proven infection [13]. In our analysis, the C-RP value at day 3 was statistically associated with ADE. Taking note of C-RP levels could reduce the length of treatment with antibiotics, but as an indicator, the C-RP level has poor specificity and low diagnostic accuracy. It cannot reliably distinguish infectious from non-infectious processes, and it is not a predictor of mortality [14,15]. Since the significance of PCT levels has not been systematically assessed among patients, we did not include them in the analysis. To suggest the timing of ADE to physicians, serial determination of PCT levels will be more useful than a single determination. However, PCT values during the first five days were not associated with survival in 48 patients with sepsis, suggesting that C-RP and PCT are not reliable markers of prognosis and should not be independently considered for predicting outcomes. 

From another perspective, in our EMW, serial determination of PCT levels was not systematically assessed in 30% of patients who de-escalated; rather, the decision to proceed with ADE was a composite decision based on multiple factors, mainly the clinical stability of patients. Thus, as previously reported [16], this could explain the higher rate of ADE on day 5 (56%) compared to day 2 (21%) and day 3 (23%). The single-center nature of the study limits the generalizability of the results. Furthermore, the fact that the qSOFA scores did not differ significantly between groups, thus implying that the severity of the illness was similar, might be due to a lack of power, for the qSOFA is only based on three items. Other scores, like the classic SOFA score, might have provided a better discrimination of the severity of patients, although the added number of items makes them more suitable for the ICU than the EMW in daily practice.

Beyond the retrospective nature of this study, even if this result was influenced by several biases, namely an adjustment to the clinical course, the multivariate analysis of mortality indicated that both ADE and an appropriate empiric treatment were protective. The retrospective nature of our study did not allow us to draw any conclusions about the effectiveness of ADE. Furthermore, we restricted inclusion to patients with any BC performed and excluded those with specific infections (e.g., pneumonia). 

## 5. Conclusions

Nevertheless, despite the aforementioned limitations, ADE is a promising approach even in an EMW setting. These results could encourage the implementation of biomarker use and wiser management of antibiotic therapy.

## Figures and Tables

**Table 1 jcm-10-03285-t001:** Characteristics of patients according to de-escalation rate at day 5.

	Overall *n* = 336	De-escalation *n* = 111 *N* (%)	No De-escalation *n* = 225 *N* (%)	*p* Value
Age (years)	68 ± 15	64 ± 14	68 ± 15	0.52
Male	194 (58%)	57 (29%)	137 (71%)	0.75
Diabetes mellitus	95 (28%)	27 (28%)	68 (72%)	0.68
Solid malignancies	94 (28%)	30 (32%)	64 (68%)	0.64
Hematologic Malignancies	55 (16%)	16 (29%)	39 (71%)	0.86
Chronic renal failure	101 (30%)	27 (27%)	74 (73%)	0.38
Transplant	15 (4%)	6 (40%)	9 (60%)	0.39
COPD	71 (21%)	18 (25%)	53 (75%)	0.33
Cardiopathies	188 (56%)	58 (31%)	130 (69%)	0.72
Cirrhosis	21 (6%)	10 (48)	11 (52%)	0.07
Dialysis	9 (3%)	2 (22%)	7 (78%)	0.6
Total parenteral nutrition	5 (1%)	3 (60%)	2 (40%)	0.14
Previous antibiotic therapies (<6 months)	171 (51%)	49 (29%)	122 (71%)	0.56
Previous steroids therapy (<3 months)	77 (23%)	21 (27)	56 (73)	0.54
Previous hospitalisation (<6 months)	250 (74%)	74 (30%)	176 (70%)	0.74
Admission from home	248 (74%)	77 (31%)	171 (69%)	0.73
Admission from health-care facilities or other wards	90 (27%)	25 (28%)	65 (72%)	0.83
Central venous catheters at time of admission	125 (37%)	40 (32%)	85 (68%)	0.55
B-D-glucan (ng/mL)	83.94	48.09	100.38	0.17
Creatinine day 1 (mg/dL)	1.81	1.47	1.95	0.28
Creatinine day 3 (mg/dL)	1.71	1.42	1.83	0.46
Creatinine day 5 (mg/dL)	1.61	1.3	1.76	0.98
qSOFA ≥ 1 day 1	85 (25%)	26 (31%)	59 (69%)	0.7
qSOFA ≥ 1 day 3	52 (15%)	16 (31%)	36 (69%)	0.91
qSOFA ≥ 1 day 5	34 (10%)	11 (32%)	23 (68%)	0.98
C-RP day 1 (mg/dL)	131.95	131.47	132.15	0.77
C-RP day 3 (mg/dL)	128.09	104.04	138.29	0.01
C-RP day 5 (mg/dL)	78.02	68.76	81.76	0.22

Abbreviations: COPD, chronic obstructive pulmonary disease; qSOFA, quick sequential organ failure assessment; C-RP, C-reactive protein.

**Table 2 jcm-10-03285-t002:** Univariate analysis of mortality according to appropriate treatment and de-escalation rates.

	Overall *N* = 336	No Survivors *N* (%)	Survivors *N* (%)	*p* Value
Empiric Therapy	325 (97)	67 (95)	258 (97)	0.485
Appropriate empiric therapy	196 (58)	31 (44)	165 (62)	0.007
Appropriate target therapy	117 (35)	16 (23)	101 (37)	0.006
De-escalation (or any de-escalation)	101 (30)	11 (15)	90 (34)	0.013

**Table 3 jcm-10-03285-t003:** Multivariate analysis of factors significantly affecting mortality.

VARIABLE	OR	IC 95%
De-escalation	0.51	0.39–0.65
Appropriate targeted therapy	0.079	0.039–0.16
Appropriate empiric therapy	0.57	1.22–3.59

Abbreviations: OR, odds Ratio.

## Data Availability

The data presented in this study are available on request from thecorresponding author.

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
