# Peer review of "Antibiotic De-escalation Experience in the Setting of Emergency Department: A Retrospective, Observational Study"

_jcm, 2021, doi:10.3390/jcm10153285_

Round 1
Reviewer 1 Report
The paper is concise and well written. I addresses a controversial antimicrobial stewardship intervention, good studies in this field are still lacking.
However, the study does not bring much to the understanding of antimicrobial de-escalation (ADE). It is true, that the studies so far have not been performed in the emergency room (ER), most studies address the use of ADE in ICU, but some of them also analyze the use of ADE in the ward patients in whom antibiotic therapy is most probably started empirically in the emergency departments. The spectrum of antibiotics used in ER is very broad for mostly community-acquired infections and the rates of multiple resistant bacteria, the share of multiple antibiotic treatment is surprising as well. It seems for the reader that an effort to improve empirical antibiotic treatment would contribute more to responsible antibiotic use than ADE.
The objectives of the study are descriptive. It is certainly impossible to say anything about resistance rates, but the authors could provide some data on antibiotic use in the cohort of patients studied. Seeing that the patients who underwent ADE received less broad-spectrum antibiotics in the following months for instance would put the study in the context of responsible use of antimicrobials.
Nevertheless, the study includes more than 300 patients and has been meticulously done.
The paper is worth publication after a major revision that would include:
- Detailed description of the intervention: where did the ADE happen, who did it, what was changed to what etc – please see the comments
- Improved objectives: the authors should include the use of antibiotics in the two groups. Showing that ADE lowered broad-spectrum antibiotic use would at leat partially compensate the bias in ADE in less severely ill patients: at least the less severely ill patients were de-escalated that let to less broad-spectrum antibiotic use
Minor comments:
Abstract, line 26: here it looks that the others received ADE later on, which is very long and may be inappropriate, it differs from the text
Introduction, line 64: the reference is a systematic review od ADE in ICU, not ER
Methods, line 107: what was considered reduction? did the authors use any of the schemes for antimicrobial spectrum, published in the literature? The authors should give a more detailed information on spectrum definitions, or they should provide the patern of deescalation (what was de-escalated to what) in the suppl. Materials
Methods, line 112: the process of ADE is not well decribed. Who did it: the physicians in the EMW, physicians on other wards, where the patients were transferred after admission to EMW? An AMS team? How long is the hospital stay in EMW?
Results, line 139: all in EMW?
Lines with CRP in table 1: did ADE happen in patients that were doing better?
Results, line 171: not clear: in 44% of patients ADE was performed up do day 3 and only additional 12% later on? It looks from the abstract that ADE happened on day 5 which would be long
Results, line 174: in how many cases one of the antibiotics was discontinued and in how many the spectrum of the pivotal antibiotic was narrowed?
Results, line 175: the authors should decribe how de-escalation was performed based on bio-markes only?
Results, line 180: it looks like a bias, these patients undervent ADE!, The authors shuld provide a comment in Discussion
Results, line 187: the same as above
Discussion, line 241: this information was not shown in the Results section - that precludes discussion on the subject
Author Response
Reviewer 1#
The paper is concise and well written. I addresses a controversial antimicrobial stewardship intervention, good studies in this field are still lacking. However, the study does not bring much to the understanding of antimicrobial de-escalation (ADE). It is true, that the studies so far have not been performed in the emergency room (ER), most studies address the use of ADE in ICU, but some of them also analyze the use of ADE in the ward patients in whom antibiotic therapy is most probably started empirically in the emergency departments. The spectrum of antibiotics used in ER is very broad for mostly community-acquired infections and the rates of multiple resistant bacteria, the share of multiple antibiotic treatment is surprising as well. It seems for the reader that an effort to improve empirical antibiotic treatment would contribute more to responsible antibiotic use than ADE. The objectives of the study are descriptive. It is certainly impossible to say anything about resistance rates, but the authors could provide some data on antibiotic use in the cohort of patients studied. Seeing that the patients who underwent ADE received less broad-spectrum antibiotics in the following months for instance would put the study in the context of responsible use of antimicrobials. Nevertheless, the study includes more than 300 patients and has been meticulously done.
The paper is worth publication after a major revision that would include:
1.Detailed description of the intervention: where did the ADE happen, who did it, what was changed to what etc – please see the comments
Dear reviewer thank you for this comment that improve notably our manuscript. We try to respond in the text and also through the minor comments. ADE was retrospectively evaluated and was carried out within EMW by physicians who worked in EMW during the period of study. You can find changes highlighted in the text.
2.Improved objectives: the authors should include the use of antibiotics in the two groups. Showing that ADE lowered broad-spectrum antibiotic use would at leat partially compensate the bias in ADE in less severely ill patients: at least the less severely ill patients were de-escalated that let to less broad-spectrum antibiotic use
Dear reviewer, thank you for these comments that improves notably our manuscript. The most frequently prescribed empirical agents were ureidopenicillins (25.1%, n = 40), carbapenems (13.6%, n = 33), glycopeptides (13.7%, n = 44), fluoroquinolones (9.6%, n = 31) and third-generation cephalosporins (6.7%, n = 16). An initial regimen that combined two agents was prescribed in 54.8% (n = 184) of cases. Despite that we can not provide at the moment further analysis showing ADE lowered broad-spectrum antibiotic accordingly to the lower number of patients and the design of the study. We hope that this paper will be of interest despite this limitation.
Minor comments:
3.Abstract, line 26: here it looks that the others received ADE later on, which is very long and may be inappropriate, it differs from the text
Dear reviewer thank you for this comment. We agree with you and we have changed accordingly to your suggestions. You can find changes highlighted in the text.
4.Introduction, line 64: the reference is a systematic review od ADE in ICU, not ER
Dear reviewer thank you for this comment, we have changed the refencence according to your suggestions.
5.Methods, line 107: what was considered reduction? did the authors use any of the schemes for antimicrobial spectrum, published in the literature? The authors should give a more detailed information on spectrum definitions, or they should provide the patern of deescalation (what was de-escalated to what) in the suppl. Materials
Dear reviewer, thank you for your suggestions that improve notably our manuscript. ADE was defined in this analysis as either reduction in the number of antibiotics, reduction of the antimicrobial spectrum or targeted de-escalation according to the microbiological results and was inspired by Kollef M.H. (Kollef, M.H. What can be expected from antimicrobial de-escalation in the critically ill?. Intensive Care Med 40, 92–95 (2014). https://doi.org/10.1007/s00134-013-3154-y) work in ICU and well outlined in figure 1 here attached. Furthermore we did not use a scheme for antimicrobial spectrum.

Reviewer 2 Report
Dear Editor, dear Authors,
This single-center retrospective study of antibiotic de-escalation (ADE) in the Emergency Ward is interesting and well-written.
Although it is only a single center study, the number of included patients is quite high, and the results are in line with previously published data. Data on ADE in the Emergency Ward is scarce, so this study brings a noteworthy insight on this topic.
The introduction, M&M and results sections do not need significant changes and are well-written.
In the discussion section I would suggest insisting in a more detailed way on the limitations:
The fact that the qSOFA scores did not differ significantly between groups, thus implying that the severity of the illness was similar, might be due to a lack of power, for the qSOFA is only based on three items. Other scores like the classic SOFA score might have provided a better discrimination of the severity of patients, although the added number of items makes them more suitable for the Intensive Care department than the Emergency Ward in daily practice.
The other point to further assert in the limitations section is that the single center nature of the study limits the generalizability of the results. The frequent use of carbapenems and low use of third generation cephalosporins is surprising and possibly related to the specific recruitment of the center. Indeed, the percentage of solid malignancies and hematologic malignancies seem high as well as the use of steroid therapy in the last 3 months. More than half the patients had a previous antibiotic therapy in the last 6 months. Perhaps a quick description of the EW patient’s recruitment might be interesting.
Best regards,
Author Response
Reviewer 2#
- Dear Editor, dear Authors,This single-center retrospective study of antibiotic de-escalation (ADE) in the Emergency Ward is interesting and well-written. Although it is only a single center study, the number of included patients is quite high, and the results are in line with previously published data. Data on ADE in the Emergency Ward is scarce, so this study brings a noteworthy insight on this topic. The introduction, M&M and results sections do not need significant changes and are well-written.
Dear reviewer thank you for your comments and thoughts
- In the discussion section I would suggest insisting in a more detailed way on the limitations: the fact that the qSOFA scores did not differ significantly between groups, thus implying that the severity of the illness was similar, might be due to a lack of power, for the qSOFA is only based on three items. Other scores like the classic SOFA score might have provided a better discrimination of the severity of patients, although the added number of items makes them more suitable for the Intensive Care department than the Emergency Ward in daily practice.
Dear reviewer thank you for these comments that improve notably our manuscript. We have added these comments and limitations to the “discussion” section. You can find changes highlighted in the text
- The other point to further assert in the limitations section is that the single center nature of the study limits the generalizability of the results.
Dear reviewer thank you for these comments that improve notably our manuscript. We have added these comments and limitations to the “discussion” section. You can find changes highlighted in the text
- The frequent use of carbapenems and low use of third generation cephalosporins is surprising and possibly related to the specific recruitment of the center. Indeed, the percentage of solid malignancies and hematologic malignancies seem high as well as the use of steroid therapy in the last 3 months. More than half the patients had a previous antibiotic therapy in the last 6 months. Perhaps a quick description of the EW patient’s recruitment might be interesting.
Dear reviewer, thank you for these comments. The frequent use of carbapenems was related to the high rate of ESBL gram-negative strains within our hospital, and the use of carbapenems became even more frequent after the MERINO trial [1]. However, we can not provide a particular reason for the different incidences of comorbidities. Despite that City of Health and Science in Turin, Italy, is a referral center for onco-haematological and rheumatological diseases and these could be a reason for the high rate of admission of this population of patients.
[1] Harris PNA et al. Effect of Piperacillin-Tazobactam vs Meropenem on 30-Day Mortality for Patients With E coli or Klebsiella pneumoniae Bloodstream Infection and Ceftriaxone Resistance: A Randomized Clinical Trial. JAMA. 2018 Sep 11;320(10):984-994. doi: 10.1001/jama.2018.12163